# Maternal constipation is associated with allergic rhinitis in the offspring: A nationwide retrospective cohort study

Ming-Hung Lee[1]☯, Meng-Che Wu[2,3,4,5]☯, Yu-Hsun Wang[6], James Cheng-Chung Wei 🄳[7,8,9,10]*

1 Department of Otorhinolaryngology Head and Neck Surgery, Taichung Veterans General Hospital, Taichung, Taiwan, 2 Division of Gastroenterology, Children's Medical Center, Taichung Veterans General Hospital, Taichung, Taiwan, 3 School of Medicine, Chung Shan Medical University, Taichung, Taiwan, 4 Department of Post-Baccalaureate, Medicine College of Medicine, National Chung Hsing University, Taichung, Taiwan, 5 Pediatric Inflammatory Bowel Disease Center, Massachusetts General Hospital, Boston, MA, United States of America, 6 Department of Medical Research, Chung Shan Medical University Hospital, Taichung, Taiwan, 7 Institute of Medicine, Chung Shan Medical University, Taichung, Taiwan, 8 Department of Nursing, Chung Shan Medical University, Taichung, Taiwan, 9 Department of Allergy, Immunology & Rheumatology, Chung Shan Medical University Hospital, Taichung, Taiwan, 10 Graduate Institute of Integrated Medicine, China Medical University, Taichung, Taiwan

☯ These authors contributed equally to this work.
* jccwei@gmail.com

**Data Availability Statement:** Information is accessible through the Taiwan National Health Insurance (NHI) Bureau's National Health Insurance Research Database (NHIRD). However,

## Abstract

Allergic rhinitis (AR) is a common atopic disease worldwide, and it was found that babies with constipation in their early life might have an increased risk of atopic diseases, including AR. Furthermore, recent studies also indicate that the maternal gut microbiota may influence babies. Thus, we extended the definition of early life in utero and evaluated the association between maternal constipation and the risk of AR in their babies. Using the Longitudinal Health Insurance Database, a subset of Taiwan's National Health Insurance Research Database, we identified 102,820 constipated mothers and 102,820 matched controls between 2005 and 2015. Propensity score analysis was used to match birth year, child sex, birth weight, gestational age, mode of delivery, maternal comorbidities, and children antibiotics taken. Multiple Cox regression and subgroup analyzes were conducted to estimate the adjusted hazard ratio of childhood AR. The incidence of childhood AR was 83.47 per 1,000 person-years in constipated mothers. Adjusting children's sex, birth weight, gestational age, mode of delivery, maternal comorbidities, and children antibiotic use, the results showed that the children whose mothers had constipation had a 1.20-fold risk of AR compared to children of mothers without constipation. Maternal constipation was associated with an increased risk of AR. Therefore, it is important to pay close attention to pregnant mothers with constipation.

## Introduction

Allergic rhinitis (AR), or allergic rhinosinusitis, is a common disease which affects around 10 to 30 percent of adults and up to 40 percent in children [1, 2]. AR can occur alongside other

adherence to the "Personal Information Protection Act" legally enforced by the Taiwanese government prohibits the public sharing of this data. For those interested in obtaining the data, the appropriate procedure involves submitting a formal proposal to the NHIRD via their website: http://nhird.nhri.org. tw. Researchers who are interested must possess valid Institutional Review Board documentation and must submit an application to the NHIRD. After undergoing a review process, there is a fee associated with obtaining database access rights. This ensures compliance with regulatory and ethical standards while facilitating the acquisition of valuable data for research purposes.

**Funding:** The author(s) received no specific funding for this work.

**Competing interests:** The authors have declared that no competing interests exist.

disorders, such as allergic conjunctivitis, sinusitis, asthma, and atopic dermatitis. While these annoying symptoms are not fatal, they frequently have a negative impact on daily life [3, 4]. When exposed to an allergen for the first time, antigen presenting cells present the allergen to CD4 T lymphocyte, which then release cytokines such as interleukin bringing about the following proinflammatory processes. The allergic-specific immunoglobulin (IgE) antibodies are produced, which bind to IgE receptors on mast cells in the mucosa over respiratory tract and basophils in peripheral blood. If the affected individual comes into contact with the same allergen again, the IgE antibodies on the mast cell will conjugate with the allergen, which then stimulates the mast cells to release granule-associated chemical mediators, resulting in the symptoms of AR [5–7]. Several risk factors have been identified or proposed for AR, including genetic inheritance or environmental exposures [8, 9], but the populations at risk of developing into AR is a matter of ongoing research. Among these, microbiota hypothesis is increasingly being discussed in the literature [10–12].

The human intestine, nose, skin, and other areas are home to the largest microbial community in the human body, composed of trillions of microbes called microbiota. Studies reported that nasal cavity harbors commensals in healthy humans, and the composition may also change in different age group. The interaction of nasal microbiome and allergen as well as microorganisms in airflow from the external environment may modulate the immune system [13–15]. Gut microbiota has been researched widely, which not only influences digestive symptoms, but has also been found to assist in the development of the immune system and has been shown to affect the balance between pro- and anti-inflammatory mechanisms [10, 16, 17]. Alterations in the gut microbiome, a condition known as gut dysbiosis, can lead to dysregulation of immune system maturation, especially in early life [10].

Most relevant research has focused on the postnatal period, because it was believed that newborn babies contact microorganisms either through the vagina during natural birth or through Cesarean section for the first time after birth. Additionally, it has been observed that infants born through Cesarean section tend to have a more favorable gut microbiota compared to those born through natural delivery. Furthermore, infants who are breastfed exhibit a more beneficial gut microbiota composition than those supplemented with formula. The use of antibiotics in infants has also been associated with a reduction in specific gut microbiota [18]. Novel concepts have been explored in recent years, such as the notion that the maternal gut microbiome promotes fetal immune system development via metabolic products during pregnancy [19, 20], and the microbiota of the mother can be transmitted to the fetus through various routes, including the placenta, and amniotic fluid [21–23]. That means the first interactions between the microbiota and the host may be initiated *in utero* [24].

Pregnant women often experience constipation and abdominal bloating and many of them need to take laxatives to relieve symptoms. Constipation prevalence rates of about 13 to 40 percent during pregnancy have been reported [25–28], which is significantly higher than nonpregnant controls [27]. It is thought that increased progesterone or other hormone concentration changes may affect small bowel and colonic motility. Iron supplementation due to iron deficiency and relatively low physical activity to avoid pregnancy complications may also play a role. In addition, there has been considerable interest in the maternal gut microbiome in recent decades, and research has shown that it changes across gestational ages [23, 29]. Gut microbiota changes can cause various abdominal symptoms, including constipation. Moreover, the maternal microbiota has the potential to cause dysbiosis in children as described above.

However, analysis of the microbiome requires advanced techniques, such as 16S rRNA gene sequencing or bacterial culture, which can be prohibitively expensive when evaluating large numbers of subjects. Dysbiosis has been associated with a number of gastrointestinal

disorders, including constipation [30–33]. Several mechanisms have been proposed to explain the link between dysbiosis and constipation. For example, certain types of bacteria in the gut may produce short-chain fatty acids (SCFAs) that help to regulate bowel movements and prevent constipation. Dysbiosis may alter the production of SCFAs, leading to constipation [34, 35]. Dysbiosis may also affect the motility of the intestines, leading to slower transit times and constipation [33, 36]. Additionally, dysbiosis may alter the production of certain hormones and neurotransmitters that play a role in bowel function [34, 35, 37]. Overall, constipation is a common symptom of gut dysbiosis according to a previous investigation and can be detected more easily.

We hypothesize that maternal constipation could influence the risk of AR in offspring and, hence, we analyzed a real-world, population-based retrospective cohort from Taiwan's National Health Insurance Research Database (NHIRD).

## Methods

### Data sources

This study utilized data obtained from multiple sources, including the National Health Insurance program in Taiwan, birth certificate applications, cause of death data, and the Maternal and Child Health Database [38], which is regulated by the Health and Welfare Data Science Center (HWDC) in Taiwan. The NHIRD [39, 40] contains all outpatient and inpatient medical claims, including drug use, medical operations, procedures, and fees. The birth certificate applications recorded birth weight, gestational weeks, delivery type, live birth, stillbirth, multiple births, and the nationality of the mother. The Maternal and Child Health Database provides the de-identified number of each mother and child's de-identified number. By linking these databases, we could trace the mother's comorbidities and medications during pregnancy.

The data used in this study were subjected to de-identification procedures, ensuring that individual participant information was not accessible to the authors during or after data collection. The databases utilized in this study contained de-identified numbers for both mothers and children. As such, the authors were unable to access individual participant identification information. Additionally, this study received an exemption from obtaining informed consent from participants, as reviewed and approved by the ethical review board of The Institutional Review Board at Chung Shan Medical University Hospital (CS2-21006). The study period for data collection was conducted in July 2022.

### Study group and outcome

We designed a retrospective cohort study to evaluate the association between maternal constipation and the risk of AR in offspring. The flow chart is shown in Fig 1.

In total, 2,146,530 individuals were enrolled from the birth certificate application database from 2005 and 2015. Individuals with the mother's identification missing, nationality missing, a foreign nationality, multiple births, or stillbirth were excluded, which left 1,897,884 children, who were included in the final analysis. The exposure group was maternal constipation, which was defined as a diagnosis of constipation (ICD-9-CM: 564.0; ICD-10-CM: K59.0) during an outpatient visit $\geq 3$ times or a hospitalization $\geq 1$ time during pregnancy plus the use of laxatives (ATC code: A06A) within one years prior to giving birth. The index date was defined as the birth date. The outcome variable was identified as a diagnosis of allergic rhinitis (ICD-9-CM: 477; ICD-10-CM: J30.1-J30.9) during an outpatient visit $\geq 3$ times or a hospitalization $\geq 1$ time. Both groups were followed up until the onset of AR, death, or Dec 31, 2017, whichever occurred first.

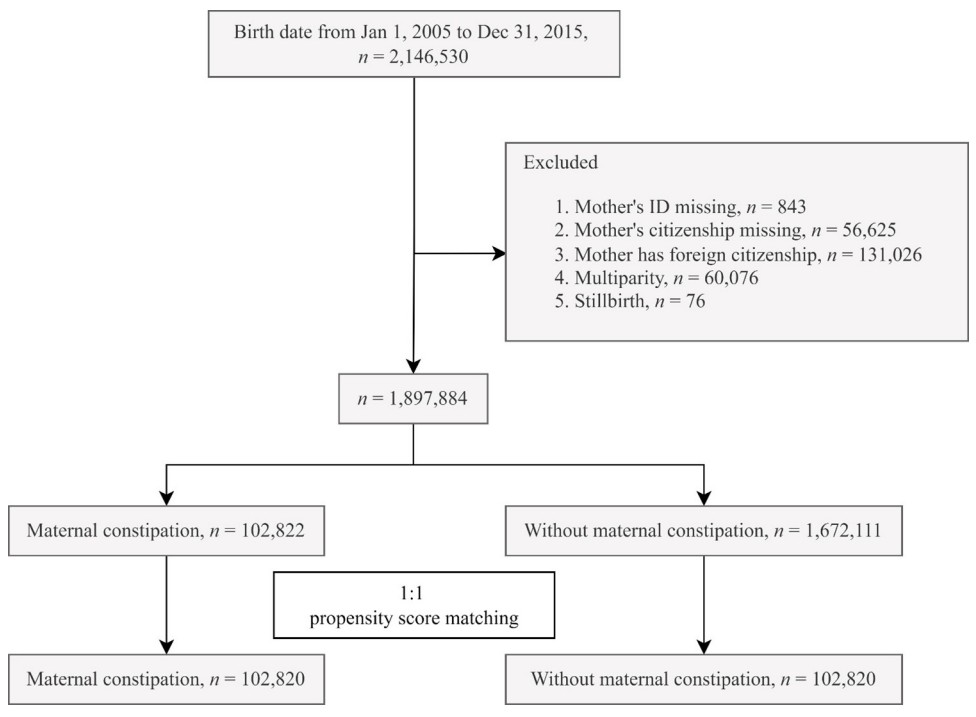

**Fig 1. Study flow chart.**

## Covariates and matching

The baseline characteristics were birth year, child's sex, birth weight (<2500; 2500–3499; ≥3500 gram), gestational age (<36; 36–40; ≥41 weeks), delivery type (normal spontaneous delivery; cesarean section), and maternal comorbidities, as follows: diabetes mellitus (ICD-9-CM: 250; ICD-10-CM: E10-14), hypertension (ICD-9-CM: 401–405; ICD-10-CM: I10-15), hyperlipidemia (ICD-9-CM: 272; ICD-10-CM: E78), gestational diabetes mellitus (ICD-9-CM: 648.8, ICD-10-CM: O99.81, O24.41–24.43), preeclampsia or eclampsia (ICD-9-CM: 642.4–642.7; ICD-10-CM: O11, O14, O15), rheumatoid arthritis (ICD-9-CM: 714.0; ICD-10-CM: M05, M06), systemic lupus erythematosus (ICD-9-CM: 710.0; ICD-10-CM: M32), Sjögren syndrome (ICD-9-CM: 710.2; ICD-10-CM: M35.0), ankylosing spondylitis (ICD-9-CM: 720.0; ICD-10-CM: M45), psoriasis (ICD-9-CM: 696.0–696.1; ICD-10-CM: L40), atopic dermatitis (ICD-9-CM: 691; ICD-10-CM: L20, L22), anxiety (ICD-9-CM: 300.0; ICD-10-CM: F41), depressive disorders (ICD-9-CM: 298.83, 296.2, 296.3, 300.4, 311; ICD-10-CM: F06.3, F32.0–32.5, F32.9, F33.0–33.4, F33.9, F34.1), asthma (ICD-9-CM: 493; ICD-10-CM: J44, J45), and allergic rhinitis (ICD-9-CM: 477; ICD-10-CM: J30.1-J30.9). These comorbidities were defined as a diagnosis within two years before the index date that was given ≥3 times or at least one hospitalization. The children use of antibiotics (ATC code: J01) set as ≥3 times prescription during the study period was recorded as well.

The propensity score matching was conducted by birth year, child's sex, birth weight, gestational weeks, delivery mode, mother's comorbidities, and antibiotic use between the two groups. The propensity score was a probability which was estimated through logistic regression, and the binary variable was the maternal constipation and non-maternal constipation group. Matching the propensity score balanced the heterogeneity of the two groups.

## Statistical analysis

To compare the maternal constipation group and non-maternal constipation groups, the absolute standardized difference (ASD) was applied. A statistical similarity between the characteristics of both groups was considered to be an ASD of less than 0.1. The relative risk (RR) and the 95% confidence intervals (C.I.) were calculated using the Poisson regression model. Kaplan-Meier analysis was applied to demonstrate the cumulative incidence of allergic rhinitis in the two groups. The log-rank test was conducted to test the significance. We performed the univariate Cox proportional hazard model to estimate the hazard ratios of the independent risk of the maternal constipation group after propensity score matching [41, 42]. The multivariate Cox proportional hazard model was used to reduce the confounding of covariates. The statistical software was SAS version 9.4 (SAS Institute, NC, USA).

# Results

A search of the database identified 2,146,530 births. After exclusion of individuals with mother's ID missing, mother's citizenship missing, foreign citizenship of mother, multiparity, and stillbirth, a total of 1,8987,884 children were left. Propensity score matching at a 1: 1 ratio was used for the maternal constipation group and the without maternal constipation group, with 102,820 children in each group. After propensity score matching, the absolute standardized differences (ASD) of all of the characteristics of children were all less than 0.1 (Table 1). The incidence of the offspring AR was 83.47 per 1,000 person-years in constipated mothers. Using Poisson regression to analyze the incidence of AR, the relative risk of the maternal constipation group was 1.19-fold that of the group without maternal constipation (relative risk: 1.19; 95% CI, 1.17–1.21) (Table 2). Kaplan-Meier analysis (Fig 2) showed higher probability of children AR statistically significant, and the log-rank test for the comparison of probability resulted in a *p* value of <0.001.

The Cox proportional hazard model analysis demonstrated that patients in the maternal constipation group had a 1.20-fold (aHR: 1.20; 95% CI, 1.18–1.22) greater risk of AR in their offspring than non-constipated mothers, with a *p* value of < 0.001 (Table 3). In addition, male children, birth weight ≥3500 gram, preterm before gestational age 36 weeks, delivery via cesarean section, mother with hyperlipidemia, Sjögren syndrome, atopic dermatitis, anxiety, asthma, and allergic rhinitis were associated with a greater risk of developing AR in children. Children who received antibiotics treatment during follow-up period has lower risk of AR (aHR: 0.43; 95% CI, 0.42–0.43). Furthermore, in the subgroup analysis, almost all of the characteristics accompanying maternal constipation showed a higher adjusted hazard ratio (Table 4). Compared to the non-constipated mothers, the constipated mothers with laxatives prescription demonstrated a higher risk of children AR regardless of the frequency of the drug use (Table 5). Follow-up of these children showed that the mean onset age of AR was 2.72 years old without maternal constipation, and 2.61 years old with maternal constipation (Table 6).

# Discussion

This is the first and largest study to investigate the association between maternal constipation and the development of AR in the mothers' offspring. To date, the majority of studies have focused on children themselves receiving agents such as antibiotics, environmental exposures, or comparisons between breastfeeding and formula feeding, as well as differences in birthing methods. These exposures have the potential to increase the risk of dysbiosis in the gut microbiota, which may eventually cause dysbiosis, resulting in allergic diseases [43–45]. Dysbiosis in the early life of children is thought to raise the risks of developing AR according to recent

**Table 1. Demographic characteristics of maternal constipation group and non-constipation group.**

| | Before PSM | | | After PSM | | |
|---|---|---|---|---|---|---|
| | Without maternal constipation N = 1672111 | Maternal constipation N = 102822 | ASD | Without maternal constipation N = 102820 | Maternal constipation N = 102820 | ASD |
| Birth year | | | 0.4492 | | | <0.001 |
| 2005 | 161664 (9.67) | 3980 (3.87) | | 3971 (3.86) | 3980 (3.87) | |
| 2006 | 159336 (9.53) | 5072 (4.93) | | 5068 (4.93) | 5072 (4.93) | |
| 2007 | 159837 (9.56) | 6264 (6.09) | | 6247 (6.08) | 6264 (6.09) | |
| 2008 | 153675 (9.19) | 7428 (7.22) | | 7425 (7.22) | 7428 (7.22) | |
| 2009 | 149478 (8.94) | 8293 (8.07) | | 8258 (8.03) | 8293 (8.07) | |
| 2010 | 118874 (7.11) | 7541 (7.33) | | 7535 (7.33) | 7541 (7.33) | |
| 2011 | 143041 (8.55) | 9980 (9.71) | | 9988 (9.71) | 9980 (9.71) | |
| 2012 | 159832 (9.56) | 12812 (12.46) | | 12815 (12.46) | 12812 (12.46) | |
| 2013 | 148336 (8.87) | 12516 (12.17) | | 12522 (12.18) | 12515 (12.17) | |
| 2014 | 153856 (9.20) | 13255 (12.89) | | 13275 (12.91) | 13255 (12.89) | |
| 2015 | 164182 (9.82) | 15681 (15.25) | | 15716 (15.28) | 15680 (15.25) | |
| Child's sex | | | 0.0089 | | | 0.0007 |
| Female | 801063 (47.91) | 49716 (48.35) | | 49751 (48.39) | 49714 (48.35) | |
| Male | 871048 (52.09) | 53106 (51.65) | | 53069 (51.61) | 53106 (51.65) | |
| Birth weight (gram) | | | 0.0564 | | | <0.001 |
| <2500 | 104954 (6.28) | 6467 (6.29) | | 6461 (6.28) | 6466 (6.29) | |
| 2500–3499 | 1305787 (78.09) | 82022 (79.77) | | 82021 (79.77) | 82021 (79.77) | |
| ≥3500 | 261370 (15.63) | 14333 (13.94) | | 14338 (13.94) | 14333 (13.94) | |
| Gestational weeks | | | 0.0641 | | | <0.001 |
| <36 weeks | 54134 (3.24) | 3188 (3.10) | | 3147 (3.06) | 3188 (3.10) | |
| 36–40 weeks | 1566574 (93.69) | 97332 (94.66) | | 97346 (94.68) | 97330 (94.66) | |
| ≥41 weeks | 51403 (3.07) | 2302 (2.24) | | 2327 (2.26) | 2302 (2.24) | |
| Delivery | | | 0.0787 | | | 0.0012 |
| NSD[1] | 1091905 (65.30) | 63249 (61.51) | | 63308 (61.57) | 63249 (61.51) | |
| C/S[2] | 580206 (34.70) | 39573 (38.49) | | 39512 (38.43) | 39571 (38.49) | |
| Maternal comorbidity | | | | | | |
| Diabetes | 15030 (0.90) | 1154 (1.12) | 0.0223 | 1095 (1.06) | 1154 (1.12) | 0.0055 |
| Hypertension | 14145 (0.85) | 779 (0.76) | 0.0099 | 703 (0.68) | 779 (0.76) | 0.0087 |
| Hyperlipidemia | 8814 (0.53) | 825 (0.80) | 0.0339 | 756 (0.74) | 824 (0.80) | 0.0076 |
| Gestational diabetes | 26032 (1.56) | 1658 (1.61) | 0.0045 | 1631 (1.59) | 1658 (1.61) | 0.0021 |
| Gestational hypertension | 5211 (0.31) | 308 (0.30) | 0.0022 | 266 (0.26) | 308 (0.30) | 0.0077 |
| Preeclampsia or eclampsia | 11434 (0.68) | 612 (0.60) | 0.0111 | 569 (0.55) | 612 (0.60) | 0.0055 |
| Rheumatoid arthritis | 1455 (0.09) | 116 (0.11) | 0.0082 | 96 (0.09) | 116 (0.11) | 0.0061 |
| Systemic lupus erythematosus | 2722 (0.16) | 198 (0.19) | 0.0071 | 196 (0.19) | 198 (0.19) | 0.0004 |
| Sjögren syndrome | 3219 (0.19) | 368 (0.36) | 0.0316 | 364 (0.35) | 367 (0.36) | 0.0005 |
| Ankylosing spondylitis | 830 (0.05) | 71 (0.07) | 0.0080 | 62 (0.06) | 71 (0.07) | 0.0034 |
| Psoriasis | 2456 (0.15) | 159 (0.15) | 0.0020 | 148 (0.14) | 159 (0.15) | 0.0028 |
| Atopic dermatitis | 15094 (0.90) | 1583 (1.54) | 0.0580 | 1543 (1.50) | 1583 (1.54) | 0.0023 |
| Anxiety | 19944 (1.19) | 3442 (3.35) | 0.1450 | 3427 (3.33) | 3440 (3.35) | 0.0007 |
| Depressive disorders | 16128 (0.96) | 3041 (2.96) | 0.1441 | 3010 (2.93) | 3039 (2.96) | 0.0017 |
| Asthma | 19169 (1.15) | 1903 (1.85) | 0.0580 | 1869 (1.82) | 1901 (1.85) | 0.0023 |
| Allergic rhinitis | 112050 (6.70) | 11263 (10.95) | 0.1502 | 11248 (10.94) | 11261 (10.95) | 0.0005 |

*(Continued)*

**Table 1.** (Continued)

| | Before PSM | | | After PSM | | |
|---|---|---|---|---|---|---|
| | **Without maternal constipation N = 1672111** | **Maternal constipation N = 102822** | **ASD** | **Without maternal constipation N = 102820** | **Maternal constipation N = 102820** | **ASD** |
| Antibiotic[3] | 1018684 (60.92) | 61721 (60.02) | 0.0183 | 61707 (60.01) | 61719 (60.03) | 0.0002 |

[1] NSD, Normal spontaneous delivery

[2] C/S, Cesarean section

[3] Children antibiotic use

research [46–48]. We also demonstrated a twice greater risks of AR in constipated patients through a retrospective cohort via NHIRD in 2020 [49]; and further, we were interested in determining whether this phenomenon manifested even earlier, i.e., in the in-utero phase, and to explore the possible impact of maternal health problems [50]. Although relatively few studies have investigated the development of allergic diseases in the prenatal phase [50–54], maternal complication and material exposure in pregnant period were more valued to influence the children's allergy nowadays. Since many pregnant women and even healthy-looking women have constipation, it has been treated as a usual problem in obstetrics and internal medicine daily practice. Nevertheless, maternal gut dysregulation may affect their children by dysbiosis, which could be neglected in the past. It appears that we should pay more attention to the pregnant mother with constipation, and the microbiota between mother and child might play a role.

Our results showed that children's characteristics (Table 3), including male gender, preterm birth under 36 weeks, and birth via cesarean section, were associated with a higher hazard ratio significantly, and these findings were similar to previous studies [55, 56]. Crump et al [57] demonstrated that low gestational age at birth was associated with a reduction in the risk of nasal corticosteroid and oral antihistamine prescription in young adulthood. It could be explained through the hygiene hypothesis proposed by Dr. Strachan in 1989 postulated that early exposure to pathogens protects against atopy. However, due to the improvement of medical facilities and hygiene of environment, we could keep opportunistic infection away from the preterm baby as possible, so further studies may be needed for preterm. When maternal comorbidities like hyperlipidemia, Sjögren's syndrome, atopic dermatitis, anxiety, asthma, or AR were concurrent, there was a notable elevation in the likelihood of AR occurrence in the offspring. As for other autoimmune diseases such as rheumatoid arthritis, systemic lupus erythematosus, ankylosing spondylitis, or psoriasis, while not statistically significant, there was still a trend towards an increased risk. According to our previous article [49], patients with hyperlipidemia, Sjögren syndrome also had higher risks of AR. Sheha et al [58, 59] found that dyslipidemia was prevalent in 56% of AR patient in their study, and higher serum total IgE, IL-

**Table 2. Poisson regression in incidence of allergic rhinitis.**

| | Without maternal constipation | Maternal constipation |
|---|---|---|
| Follow-up duration (years) | 482187 | 458351 |
| Number of allergic rhinitis | 33848 | 38257 |
| Incidence (95% C.I.)[1] | 70.19 (69.45–70.95) | 83.47 (82.64–84.31) |
| Relative risk (95% C.I.) | Reference | 1.19 (1.17–1.21) |

[1] per 1000 years

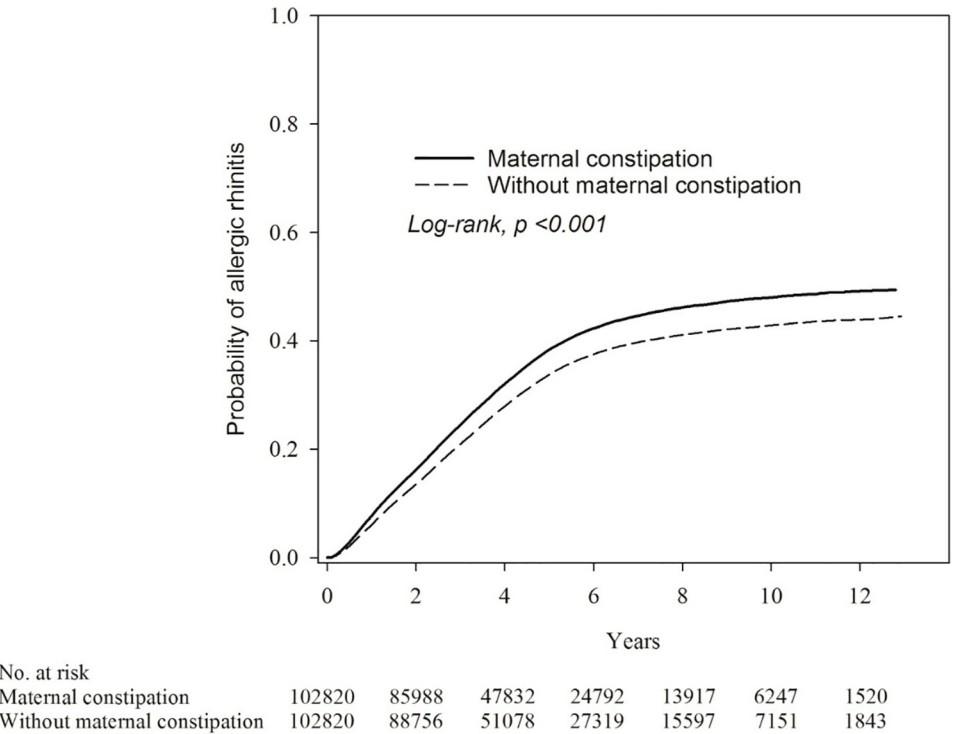

**Fig 2. Kaplan-Meier curve of probability of AR in maternal constipation and maternal non-constipation groups.**

17A level were independent risk factors for dyslipidemia among AR patients. A positive correlation was found between total cholesterol and the severity of AR [59], and Manti et al [60] also presented similar results. Past research has examined the correlation between several immune diseases and AR, yielding varying results. While divergent findings existed, a substantial portion of studies had reported a positive association [61–63]. Several previous studies have indicated that in mothers with atopic disease, environmental exposure during pregnancy and pregnancy complications were associated with child atopic disease [54, 64, 65]. Antibiotics exposure during the follow up period in our investigating group of children had a relatively low hazard ratio among all of the characteristics, with an up to 0.43-fold difference compared with the control group. A few studies [46, 66–68] have focused on exposure to antibiotics and association to atopic disease in early life, and controversial findings were shown. Liu et al [69] conducted a meta-analysis to investigate the correlations and the incidence of AR was higher in antibiotics group, but they found most included studies analysis of exposure to antibiotic as a dichotomous variable without information on the type and dose of antibiotics.

In the subgroup analysis (Table 4), mothers with constipation had a significantly higher hazard ratio for AR in their offspring regardless of the child's sex, birth weight, gestational age, and delivery mode, as well as maternal hypertension, diabetes, gestational diabetes, hyperlipidemia, preeclampsia/eclampsia, atopic dermatitis, depressive disorder, or children antibiotics use. In the context of maternal comorbidities, if the mother had hypertension, hyperlipidemia, preeclampsia, eclampsia, or psoriasis, and simultaneously experienced constipation, it significantly increased the risk of her offspring developing AR. On the other hand, if the mother had autoimmune diseases such as rheumatoid arthritis, systemic lupus erythematosus, Sjögren syndrome, or ankylosing spondylitis, and also experienced constipation, the risk of her offspring developing AR was comparatively less significant. This may be because having autoimmune

**Table 3. Cox proportional hazard model analysis for risk of allergic rhinitis in offspring.**

|  | Crude HR | p value | Adjusted HR | p value |
|---|---|---|---|---|
| Maternal constipation |  |  |  |  |
| No | Reference |  | Reference |  |
| Yes | 1.18 (1.16–1.20) | <0.001 | 1.20 (1.18–1.22) | <0.001 |
| Child's sex |  |  |  |  |
| Female | Reference |  | Reference |  |
| Male | 1.26 (1.24–1.28) | <0.001 | 1.28 (1.26–1.30) | <0.001 |
| Birth weight (gram) |  |  |  |  |
| 2500–3499 | Reference |  | Reference |  |
| <2500 | 1.01 (0.98–1.04) | 0.4363 | 0.99 (0.95–1.02) | 0.3794 |
| ≥3500 | 1.05 (1.03–1.07) | <0.001 | 1.03 (1.01–1.05) | 0.0069 |
| Gestational weeks |  |  |  |  |
| 36–40 weeks | Reference |  | Reference |  |
| <36 weeks | 1.12 (1.07–1.17) | <0.001 | 1.13 (1.08–1.19) | <0.001 |
| ≥41 weeks | 0.99 (0.95–1.04) | 0.6985 | 1.03 (0.98–1.08) | 0.288 |
| Delivery |  |  |  |  |
| NSD[1] | Reference |  | Reference |  |
| C/S[2] | 1.05 (1.04–1.07) | <0.001 | 1.07 (1.05–1.08) | <0.001 |
| Maternal comorbidity |  |  |  |  |
| Diabetes | 1.07 (1.00–1.15) | 0.0476 | 1.03 (0.96–1.10) | 0.465 |
| Hypertension | 1.01 (0.93–1.10) | 0.8014 | 0.94 (0.86–1.03) | 0.1814 |
| Hyperlipidemia | 1.24 (1.15–1.34) | <0.001 | 1.19 (1.09–1.29) | <0.001 |
| Gestational diabetes | 1.05 (0.99–1.11) | 0.1269 | 1.01 (0.95–1.07) | 0.725 |
| Gestational hypertension | 0.98 (0.85–1.13) | 0.764 | 0.95 (0.82–1.09) | 0.4418 |
| Preeclampsia or eclampsia | 0.96 (0.87–1.06) | 0.4612 | 0.96 (0.87–1.06) | 0.3926 |
| Rheumatoid arthritis | 1.13 (0.91–1.40) | 0.2578 | 1.12 (0.90–1.39) | 0.3087 |
| Systemic lupus erythematosus | 1.22 (1.05–1.43) | 0.0113 | 1.12 (0.96–1.32) | 0.1498 |
| Sjögren syndrome | 1.27 (1.13–1.42) | <0.001 | 1.16 (1.03–1.30) | 0.013 |
| Ankylosing spondylitis | 1.04 (0.78–1.39) | 0.7701 | 1.05 (0.79–1.40) | 0.7263 |
| Psoriasis | 1.15 (0.95–1.38) | 0.1549 | 1.12 (0.93–1.35) | 0.2529 |
| Atopic dermatitis | 1.12 (1.06–1.19) | <0.001 | 1.08 (1.02–1.15) | 0.0057 |
| Anxiety | 1.12 (1.08–1.17) | <0.001 | 1.07 (1.03–1.12) | 0.0014 |
| Depressive disorders | 1.08 (1.04–1.13) | 0.0002 | 1.03 (0.99–1.08) | 0.1632 |
| Asthma | 1.40 (1.33–1.47) | <0.001 | 1.20 (1.14–1.26) | <0.001 |
| Allergic rhinitis | 1.52 (1.49–1.55) | <0.001 | 1.50 (1.47–1.54) | <0.001 |
| Antibiotic[3] | 0.44 (0.43–0.44) | <0.001 | 0.43 (0.42–0.43) | <0.001 |

[1] NSD, Normal spontaneous delivery

[2] C/S, Cesarean section

[3] Children antibiotics use.

conditions in itself could lead to AR in the offspring [70, 71], and thus reducing the relative contribution of constipation to AR. Our previous population-based retrospective cohort study investigating the correlation between constipation and AR revealed similar results in the subgroup analysis involving autoimmune diseases [49]. Although some factors did not reach statistical significance, it remains noteworthy that the presence of maternal comorbidities combined with constipation tends to elevate the risk of AR. However, it is important to note that the number of mothers with autoimmune diseases in our study was very small, so it may not accurately reflect the precise risk.

**Table 4. Subgroup analysis of the association between maternal constipation and development of AR in offspring.**

| | Number of allergic rhinitis | | | |
| | Without maternal constipation | Maternal constipation | Adjusted HR | p value |
|---|---|---|---|---|
| Child's sex | | | | |
| Female | 14909 (29.97) | 16884 (33.96) | 1.20 (1.17–1.22) | <0.001 |
| Male | 18939 (35.69) | 21373 (40.25) | 1.20 (1.18–1.23) | <0.001 |
| | | | | p for interaction = 0.6334 |
| Birth weight (gram) | | | | |
| <2500 | 2073 (32.08) | 2368 (36.62) | 1.22 (1.15–1.29) | <0.001 |
| 2500–3499 | 26817 (32.7) | 30306 (36.95) | 1.20 (1.18–1.22) | <0.001 |
| ≥3500 | 4958 (34.58) | 5583 (38.95) | 1.20 (1.15–1.24) | <0.001 |
| | | | | p for interaction = 0.8887 |
| Gestational weeks | | | | |
| <36 weeks | 1098 (34.89) | 1243 (38.99) | 1.21 (1.12–1.31) | <0.001 |
| 36–40 weeks | 31921 (32.79) | 36109 (37.1) | 1.20 (1.18–1.22) | <0.001 |
| ≥41 weeks | 829 (35.63) | 905 (39.31) | 1.17 (1.06–1.28) | 0.0012 |
| | | | | p for interaction = 0.7369 |
| Delivery | | | | |
| NSD[1] | 20602 (32.54) | 23140 (36.59) | 1.19 (1.17–1.21) | <0.001 |
| C/S[2] | 13246 (33.52) | 15117 (38.2) | 1.22 (1.19–1.25) | <0.001 |
| | | | | p for interaction = 0.1679 |
| Diabetes | | | | |
| No | 33474 (32.91) | 37817 (37.2) | 1.20 (1.18–1.22) | <0.001 |
| Yes | 374 (34.16) | 440 (38.13) | 1.21 (1.05–1.39) | 0.0089 |
| | | | | p for interaction = 0.8798 |
| Hypertension | | | | |
| No | 33633 (32.94) | 37964 (37.2) | 1.20 (1.18–1.22) | <0.001 |
| Yes | 215 (30.58) | 293 (37.61) | 1.28 (1.07–1.53) | 0.0065 |
| | | | | p for interaction = 0.5185 |
| Hyperlipidemia | | | | |
| No | 33579 (32.9) | 37896 (37.15) | 1.20 (1.18–1.22) | <0.001 |
| Yes | 269 (35.58) | 361 (43.81) | 1.25 (1.07–1.47) | 0.006 |
| | | | | p for interaction = 0.4411 |
| Gestational diabetes | | | | |
| No | 33306 (32.91) | 37661 (37.23) | 1.20 (1.18–1.22) | <0.001 |
| Yes | 542 (33.23) | 596 (35.95) | 1.14 (1.02–1.28) | 0.0266 |
| | | | | p for interaction = 0.4617 |
| Gestational hypertension | | | | |
| No | 33765 (32.92) | 38150 (37.22) | 1.20 (1.18–1.22) | <0.001 |
| Yes | 83 (31.2) | 107 (34.74) | 1.22 (0.91–1.64) | 0.1865 |
| | | | | p for interaction = 0.9920 |
| Preeclampsia or eclampsia | | | | |
| No | 33675 (32.93) | 38037 (37.22) | 1.20 (1.18–1.22) | <0.001 |
| Yes | 173 (30.4) | 220 (35.95) | 1.24 (1.01–1.52) | 0.0384 |
| | | | | p for interaction = 0.7204 |
| Rheumatoid arthritis | | | | |
| No | 33811 (32.91) | 38211 (37.2) | 1.20 (1.18–1.22) | <0.001 |
| Yes | 37 (38.54) | 46 (39.66) | 1.03 (0.63–1.67) | 0.9219 |
| | | | | p for interaction = 0.5741 |

*(Continued)*

**Table 4.** (Continued)

| | Number of allergic rhinitis | | | |
| | Without maternal constipation | Maternal constipation | Adjusted HR | p value |
|---|---|---|---|---|
| Systemic lupus erythematosus | | | | |
| No | 33768 (32.9) | 38178 (37.2) | 1.20 (1.18–1.22) | <0.001 |
| Yes | 80 (40.82) | 79 (39.9) | 0.96 (0.70–1.33) | 0.8195 |
| | | | | p for interaction = 0.2072 |
| Sjögren syndrome | | | | |
| No | 33707 (32.9) | 38099 (37.19) | 1.20 (1.18–1.22) | <0.001 |
| Yes | 141 (38.74) | 158 (43.05) | 1.07 (0.85–1.34) | 0.5915 |
| | | | | p for interaction = 0.4313 |
| Ankylosing spondylitis | | | | |
| No | 33828 (32.92) | 38230 (37.21) | 1.20 (1.18–1.22) | <0.001 |
| Yes | 20 (32.26) | 27 (38.03) | 1.23 (0.65–2.35) | 0.523 |
| | | | | p for interaction = 0.6150 |
| Psoriasis | | | | |
| No | 33804 (32.92) | 38192 (37.2) | 1.20 (1.18–1.22) | <0.001 |
| Yes | 44 (29.73) | 65 (40.88) | 1.61 (1.08–2.41) | 0.0196 |
| | | | | p for interaction = 0.2642 |
| Atopic dermatitis | | | | |
| No | 33289 (32.87) | 37615 (37.16) | 1.20 (1.18–1.22) | <0.001 |
| Yes | 559 (36.23) | 642 (40.56) | 1.15 (1.02–1.29) | 0.0194 |
| | | | | p for interaction = 0.4921 |
| Anxiety | | | | |
| No | 32635 (32.83) | 36911 (37.14) | 1.20 (1.18–1.22) | <0.001 |
| Yes | 1213 (35.4) | 1346 (39.13) | 1.19 (1.10–1.29) | <0.001 |
| | | | | p for interaction = 0.7384 |
| Depressive disorders | | | | |
| No | 32781 (32.84) | 37083 (37.16) | 1.20 (1.18–1.22) | <0.001 |
| Yes | 1067 (35.45) | 1174 (38.63) | 1.14 (1.05–1.24) | 0.0015 |
| | | | | p for interaction = 0.2627 |
| Asthma | | | | |
| No | 33053 (32.74) | 37371 (37.03) | 1.20 (1.18–1.22) | <0.001 |
| Yes | 795 (42.54) | 886 (46.61) | 1.15 (1.04–1.26) | 0.0051 |
| | | | | p for interaction = 0.3848 |
| Allergic rhinitis | | | | |
| No | 28936 (31.6) | 33049 (36.1) | 1.21 (1.20–1.23) | <0.001 |
| Yes | 4912 (43.67) | 5208 (46.25) | 1.12 (1.07–1.16) | <0.001 |
| | | | | p for interaction <0.001 |
| Antibiotic[1] | | | | |
| No | 16491 (40.11) | 18922 (46.04) | 1.24 (1.21–1.26) | <0.001 |
| Yes | 17357 (28.13) | 19335 (31.33) | 1.15 (1.12–1.17) | <0.001 |
| | | | | p for interaction <0.001 |

[1]Children antibiotic use.

With regards to laxatives effect, the children whose mothers took laxatives during pregnancy had a higher risk of AR comparing to the non-constipated group generally (Table 5), but the dose effect of the laxatives still remained unclear. Whether the more laxatives use

**Table 5. Cox proportional hazard model analysis for risk of AR in laxatives frequency.**

| | Number of children with AR (Total children number) | Crude HR | Adjusted HR |
|---|---|---|---|
| Non-constipation group | 33848(102820) | Reference | Reference |
| Maternal constipation with laxatives prescription <3 times | 18239(46649) | 1.21 (1.19–1.24) * | 1.24 (1.22–1.27) * |
| Maternal constipation with laxatives prescription ≥3 times | 20018(56171) | 1.15 (1.13–1.17) * | 1.16 (1.14–1.18) * |
| Maternal constipation with laxatives prescription <6 times | 31874(84229) | 1.19 (1.17–1.21) * | 1.22 (1.20–1.24) * |
| Maternal constipation with laxatives prescription ≥6 times | 6383(18591) | 1.12 (1.09–1.15) * | 1.12 (1.10–1.16) * |

*$p$ value < 0.01

would decrease the risk of developing offspring AR still needs to be investigated. Further prospective or retrospective cohort studies to determine the relationship between laxatives use and the risk of AR in offspring could be considered in the future. It is also worth noting that our result revealed the children of maternal constipation group had earlier onset to be diagnosed with AR significantly (Table 6), which indicated that dysbiosis might be an accelerative factor in early life.

Although the transmission of gut microbiota from mother to child has still not been fully elucidated, a number of factors are considered important determinants, such as the mothers' genotypes, diet, mode of delivery, and breastfeeding, and any of these may disrupt the transmission [72]. Additionally, more and more evidence [21–24] showed microorganisms from mothers may transmit to fetus via placenta and amniotic fluid. These microorganisms may be colonized in the fetus body; likewise, the maternal microbiota seems to be more persistent in the infant gut [73]. Shilts et al [74] sampled the nasal cavity of the infant after birth and found microbiome as well, and the microbiota was composed of maternal vaginal and skin bacteria predominantly, and also other genera which might be transmitted during pregnancy as well. Rinninella et al [75] studied the healthy microbiota composition in human and proposed the richer and more diverse the microbiota, the better ability to withstand external threat. However, if maternal dysbiosis happens first, microbiota might not be transmitted to the child, even if the pathway is not disrupted, and eventually decreases the diversity of the microbiome in babies.

The main strengths of our study are its large sample size and the relatively long follow-up period. The integrated information collected during medical services was accessible for every case. Thus, the selection bias, information bias, and recall bias were probably relatively low. Indeed, there were still several limitations in this study. Firstly, we acknowledge that our current study is retrospective. To enhance the robustness of our findings, prospective cohort studies or even randomized control trials, although challenging in pregnant populations, could

**Table 6. Follow up period duration and time to AR onset.**

| | Number of children | Mean(years) | SD[1](years) | $p$ value |
|---|---|---|---|---|
| Follow-up duration, years | | | | <0.001 |
| Without maternal constipation | 102820 | 4.69 | 2.91 | |
| Maternal constipation | 102820 | 4.46 | 2.86 | |
| Time to AR onset, years | | | | <0.001 |
| Without maternal constipation | 33848 | 2.72 | 1.82 | |
| Maternal constipation | 38257 | 2.61 | 1.81 | |

[1]SD: standard deviation

provide higher-level evidence to substantiate our hypotheses. Secondly, medical doctors may give the diagnosis depending on his or her experience and expertise, and Taiwan's National Health Insurance administration examines claims data to prevent violations. Additionally, the NHIRD does not include data on covariates, including family history, genetic data, social adversity, laboratory data, and some environmental exposures. Despite the use of propensity score matching for selected comorbidities and individual characteristics, any unmeasured confounding factors might have affected the results. Additionally, the diagnoses of AR and constipation were based on the ICD-9-CM and ICD-10-CM codes in the dataset, and there was no specific IgE immunoassays, skin prick tests, 16S rRNA sequencing or fecal bacterial culture available in the data. Since we did not individually and comprehensively review every patient's medical record, the accuracy of the diagnoses could not be confirmed. Besides, exploring the microbiomics and metabolomics aspects, potentially through fecal microbiota analysis in pregnant women with constipation, could provide valuable insights into the role of gut microbiota.

## Conclusion

To conclude, children born to mothers who were constipated during pregnancy had a 1.20-fold greater risk for AR compared with those born to non-constipated mothers (aHR: 1.20; 95% CI, 1.18–1.22). The use of laxatives during pregnancy in constipated mothers also elevated the risks for AR in their children. The pathophysiological association between maternal constipation and offspring AR requires further research.

## Supporting information

**S1 Checklist. STROBE statement—checklist of items that should be included in reports of observational studies.**
(DOCX)

## Acknowledgments

This study is based in part on data from the National Health Insurance Research Database provided by the Bureau of National Health Insurance, Department of Health and managed by National Health Research Institutes. The interpretation and conclusions contained herein do not represent those of National Health Insurance Administration, Department of Health or National Health Research Institutes.

## Author Contributions

**Conceptualization:** Ming-Hung Lee, Meng-Che Wu.

**Data curation:** Yu-Hsun Wang.

**Formal analysis:** Meng-Che Wu, Yu-Hsun Wang.

**Investigation:** Ming-Hung Lee, Meng-Che Wu, James Cheng-Chung Wei.

**Methodology:** Meng-Che Wu, Yu-Hsun Wang.

**Project administration:** James Cheng-Chung Wei.

**Resources:** Meng-Che Wu, James Cheng-Chung Wei.

**Software:** Yu-Hsun Wang.

**Supervision:** Meng-Che Wu, James Cheng-Chung Wei.

**Validation:** Meng-Che Wu.

**Visualization:** Ming-Hung Lee, Yu-Hsun Wang.

**Writing – original draft:** Ming-Hung Lee, Meng-Che Wu.

**Writing – review & editing:** Ming-Hung Lee, Meng-Che Wu.

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
