## [Decision Letter · Decision Letter 0]

31 Jul 2023

PONE-D-23-15200Maternal Constipation is Associated with Allergic Rhinitis in the Offspring: A Nationwide Retrospective Cohort StudyPLOS ONE

Dear Dr. Wei,

Thank you for submitting your manuscript to PLOS ONE. After careful consideration, we feel that it has merit but does not fully meet PLOS ONE’s publication criteria as it currently stands. Therefore, we invite you to submit a revised version of the manuscript that addresses the points raised during the review process.

We look forward to receiving your revised manuscript.

Kind regards,

Dong Keon Yon, MD, FACAAI, FAAAAI

Academic Editor

PLOS ONE

Additional Editor Comments:

Thank you for submitting your manuscript. The reviewers and I believe it is of potential value for our readers. However, the reviewers have raised a number of very important issues, and their excellent comments will need to be adequately addressed in a revision before the acceptability of your manuscript for publication in the Journal can be determined. We cannot guarantee that your revised paper will be chosen for publication; this would be solely based on how satisfactorily you have addressed the reviewer comments.

#1. Kaplan-Meier analysis was applied to demonstrate the cumulative incidence of allergic rhinitis in the two groups.-> Please cite the statistical guideline (DOI: https://doi.org/10.54724/lc.2023.e8).

Reviewers' comments:

Reviewer's Responses to Questions

**Comments to the Author**

1. Is the manuscript technically sound, and do the data support the conclusions?

Reviewer #1: Yes

Reviewer #2: Yes

2. Has the statistical analysis been performed appropriately and rigorously? 

Reviewer #1: Yes

Reviewer #2: Yes

3. Have the authors made all data underlying the findings in their manuscript fully available?

Reviewer #1: Yes

Reviewer #2: Yes

4. Is the manuscript presented in an intelligible fashion and written in standard English?

Reviewer #1: Yes

Reviewer #2: Yes

5. Review Comments to the Author

Reviewer #1: Lee et al. demonstrated the relationship between maternal constipation and the development of Allergic Rhinitis (AR) in the offspring. This study contributes to our understanding of the effect of maternal constipation on the risk of AR in offspring. However, there are some concerns regarding the study that need to be addressed.

1. Table 3 illustrates a significantly higher hazard ratio in males. Existing studies generally showed higher constipation prevalence in females (https://bmcgastroenterol.biomedcentral.com/articles/10.1186/s12876-020-01306-y). Hence, an in-depth discussion is required to elucidate the increased hazard ratio observed in males.

2. In Table 5, the frequency of laxative usage suggests a potential interpretation: if the frequency is interpreted as constipation severity, a higher risk of allergic rhinitis is seen in mild constipation. This finding warrants further exploration in the discussion section.

3. In the discussion about the potential influence of laxatives, you suggest that a higher dose of laxatives might decrease the risk of developing offspring AR. However, this statement seems speculative and would benefit from supporting evidence or a more in-depth discussion.

4. The section discussing the relationship between mothers with autoimmune diseases and the risk of AR in offspring is a bit unclear. The authors might consider expanding on the rationale behind why mothers with certain autoimmune diseases and constipation show a lower risk of AR in their offspring than mothers without these diseases.

Reviewer #2: Dear authors,

I have now completed the review of the manuscript titled "Maternal Constipation is Associated with Allergic Rhinitis in

the Offspring: A Nationwide Retrospective Cohort Study."

The manuscript is interesting and, in general, fair written.

I have some suggestions to further improve the quality of the manuscript.

1. The first few sections introduced some relevant articles. Please explain the results or summarize with effect sizes.

2. I suggest authors clarify how other researchers can obtain the original data.

3. Please compare the prevalence of defined diseases to the previous national or worldwide data.

4. In the ‘Statistical Analysis’ section, refer to some statistical standards and guidelines on Kaplan-Meier and Cox proportional hazards regression in survival analysis.

5. What is the future scope of the proposed research, authors have described the limitations in a good way, and I suggest that these can be the future scope of the work.

6. PLOS authors have the option to publish the peer review history of their article (what does this mean?). If published, this will include your full peer review and any attached files.

Reviewer #1: **Yes: **Jiseung Kang

Reviewer #2: No

---

## [Author Response · Author response to Decision Letter 0]

11 Sep 2023

Reviewer #1:

1. Table 3 illustrates a significantly higher hazard ratio in males. Existing studies generally showed higher constipation prevalence in females (https://bmcgastroenterol.biomedcentral.com/articles/10.1186/s12876-020-01306-y). Hence, an in-depth discussion is required to elucidate the increased hazard ratio observed in males.

Response:

 We appreciate the reviewer’s attention to detail and apologize for any confusion that may have arisen. We would like to clarify that while existing literature indeed indicates a higher prevalence of constipation among females, the data presented in Table 3 highlights a different aspect.

The information in Table 3 demonstrates a significantly higher hazard ratio for the development of allergic rhinitis (AR) in males compared to females among the offspring. This observation is distinct from the prevalence of constipation. We will ensure that this distinction is explicitly addressed and discussed in our revised manuscript to avoid any misunderstanding. 

(Please see page 10 line 200-203)

2. In Table 5, the frequency of laxative usage suggests a potential interpretation: if the frequency is interpreted as constipation severity, a higher risk of allergic rhinitis is seen in mild constipation. This finding warrants further exploration in the discussion section.

Response:

Regarding Table 5, it is important to note that when comparing mothers with constipation, regardless of whether they used laxatives, to the non-constipated group, the hazard ratios for allergic rhinitis (AR) are consistently higher. Therefore, a more fitting interpretation would be that laxatives may alleviate the symptoms of constipation, potentially leading to some degree of improvement in the dysbiosis within the body. However, we cannot assert that laxatives could reduce the risk of AR and we hadn’t found a dose-dependent effect of laxatives on AR risk based on our data. This also paves the way for future research, aiming to delve deeper into the potential effects of laxatives on risk.

3. In the discussion about the potential influence of laxatives, you suggest that a higher dose of laxatives might decrease the risk of developing offspring AR. However, this statement seems speculative and would benefit from supporting evidence or a more in-depth discussion.

Response:

Currently, there is no study available for citation that specifically explore the relationship between laxatives and AR. As pioneers in introducing this possibility, future research can delve deeper into this aspect. We acknowledge the need for further investigation to elucidate the complex relationship between laxatives and AR risk, and we appreciate the reviewer's discerning observation. Our study lays the foundation, and we look forward to the subsequent studies that may provide more comprehensive insights into this intriguing topic.

4. The section discussing the relationship between mothers with autoimmune diseases and the risk of AR in offspring is a bit unclear. The authors might consider expanding on the rationale behind why mothers with certain autoimmune diseases and constipation show a lower risk of AR in their offspring than mothers without these diseases.

Response:

 In our study, we found that if mothers had autoimmune diseases such as rheumatoid arthritis, systemic lupus erythematosus, Sjögren syndrome, or ankylosing spondylitis, and also experienced constipation, the risk of their offspring developing AR appeared comparatively less significant. This may be attributed to the fact that autoimmune conditions themselves could lead to AR in offspring, reducing the relative contribution of constipation to AR. Our previous population-based retrospective cohort study involving autoimmune diseases similarly revealed a similar trend.

It's important to note that while the presence of maternal comorbidities combined with constipation tends to elevate the risk of AR, the number of mothers with autoimmune diseases in our study was limited, which may not accurately reflect the precise risk. Furthermore, Table 4 highlights that maternal autoimmune-related comorbidities, while reducing the risk, still demonstrate an overall higher risk trend when it comes to AR in offspring. This aligns with previous research indicating a positive correlation between autoimmune diseases and AR.

In summary, it appears that within the subgroup analysis, the risk is somewhat attenuated due to the presence of autoimmune diseases, but there remains a noticeable trend indicating an increased risk of AR in offspring when mothers have autoimmune conditions alongside constipation.

(Please see page 16 line 278-291)

 

Reviewer #2:

1. The first few sections introduced some relevant articles. Please explain the results or summarize with effect sizes.

Response:

We appreciate the reviewer's suggestion. We have added additional sections to elucidate the results of the relevant articles mentioned, with specific highlights and references within the manuscript. Please refer to page 3, line 68-74 for the detailed explanations

2. I suggest authors clarify how other researchers can obtain the original data. 

Response:

 Information is accessible through the Taiwan National Health Insurance (NHI) Bureau's National Health Insurance Research Database (NHIRD). However, adherence to the "Personal Information Protection Act" legally enforced by the Taiwanese government prohibits the public sharing of this data. For those interested in obtaining the data, the appropriate procedure involves submitting a formal proposal to the NHIRD via their website: http://nhird.nhri.org.tw. Researchers who are interested must possess valid Institutional Review Board documentation and must submit an application to the NHIRD. After undergoing a review process, there is a fee associated with obtaining database access rights. This ensures compliance with regulatory and ethical standards while facilitating the acquisition of valuable data for research purposes.

3. Please compare the prevalence of defined diseases to the previous national or worldwide data.

Response:

 We appreciate the thoughtful question posed by the reviewer. It is indeed valuable to compare the prevalence of defined diseases in our study to previous national or worldwide data. However, we must acknowledge certain limitations that hindered such a direct comparison.

 Firstly, the population within our study was carefully selected based on specific criteria, which makes it challenging to calculate the precise prevalence rates of constipation among pregnant women and allergic rhinitis (AR) in children in Taiwanese population. Due to constraints related to database access, obtaining accurate nationwide prevalence rate data for Taiwan was not feasible currently. In our study, the proportion rates of constipation and AR calculated from our study population appear lower than the reported prevalence rates in previous research. This difference may be attributed to the stringent criteria we applied, possibly influenced by the context of our healthcare system under a national insurance framework. 

It's worth noting that our diagnostic criteria for AR relied on ICD diagnosis codes, specifically outpatient visits of three or more times or hospitalization at least once. Similarly, for constipation, we employed ICD diagnosis codes for outpatient visits of three or more times or hospitalization at least once during pregnancy, coupled with the use of laxatives (ATC code: A06A) within one year prior to giving birth. This approach aligns with the established methodology commonly used in research utilizing healthcare insurance databases, providing a robust and defensible definition.

In conclusion, while direct comparisons to previous national or worldwide prevalence data present challenges in our specific context, we believe that our study's methodology for defining and identifying cases remains sound and aligns with established practices in healthcare database research.

We appreciate the reviewer's insight and acknowledge that future research could explore this aspect further and delve into the reasons behind such variations in prevalence rates.

4. In the ‘Statistical Analysis’ section, refer to some statistical standards and guidelines on Kaplan-Meier and Cox proportional hazards regression in survival analysis.

Response:

 We have included additional references to statistical literature as follows and have listed them in the references section:

Lee, S. W. Kaplan-Meier and Cox proportional hazards regression in survival analysis: statistical standard and guideline of Life Cycle Committee. Life Cycle 3, e8, doi:10.54724/lc.2023.e8 (2023).

Katz MH, Hauck WW. Proportional hazards (Cox) regression. J Gen Intern Med.

1993 Dec;8(12):702-11. doi: 10.1007/BF02598295 . PMID: 8120690 .

(Please see subsection “Statistical analysis” page7-8 line 163-173)

5. What is the future scope of the proposed research, authors have described the limitations in a good way, and I suggest that these can be the future scope of the work.

Response:

In terms of future directions for our research, we acknowledge that our current study is retrospective. To enhance the robustness of our findings, prospective cohort studies or even randomized control trials, although challenging in pregnant populations, could provide higher-level evidence to substantiate our hypotheses. Additionally, considering the specific criteria used in our study's cohort selection, future research could involve accessing relevant databases to identify constipation and AR prevalence rates among pregnant women and children in Taiwan. This would allow for comparisons with prior epidemiological studies, enriching our understanding of these conditions within these populations. Furthermore, exploring the microbiomics and metabolomics aspects, potentially through fecal microbiota analysis in pregnant women with constipation, could provide valuable insights into the role of gut microbiota in this context. These avenues of basic research could offer substantial support for our arguments and further expand the scope of our investigations.

 We have addressed this part of the response by making modifications in the “Discussion” section.

(Please see page 17 line 316-331)

---

## [Decision Letter · Decision Letter 1]

26 Sep 2023

Maternal Constipation is Associated with Allergic Rhinitis in the Offspring: A Nationwide Retrospective Cohort Study

PONE-D-23-15200R1

Dear Dr. Wei,

We’re pleased to inform you that your manuscript has been judged scientifically suitable for publication and will be formally accepted for publication once it meets all outstanding technical requirements.

Kind regards,

Dong Keon Yon, MD, FACAAI, FAAAAI

Academic Editor

PLOS ONE

Additional Editor Comments (optional):

This is an excellent paper.

Reviewers' comments:

Reviewer's Responses to Questions

**Comments to the Author**

1. If the authors have adequately addressed your comments raised in a previous round of review and you feel that this manuscript is now acceptable for publication, you may indicate that here to bypass the “Comments to the Author” section, enter your conflict of interest statement in the “Confidential to Editor” section, and submit your "Accept" recommendation.

Reviewer #2: All comments have been addressed

2. Is the manuscript technically sound, and do the data support the conclusions?

Reviewer #2: Yes

3. Has the statistical analysis been performed appropriately and rigorously? 

Reviewer #2: Yes

4. Have the authors made all data underlying the findings in their manuscript fully available?

Reviewer #2: Yes

5. Is the manuscript presented in an intelligible fashion and written in standard English?

Reviewer #2: Yes

6. Review Comments to the Author

Reviewer #2: All comments have been addressed. Thank you to the authors and editors for considering my opinion on this manuscript.

7. PLOS authors have the option to publish the peer review history of their article (what does this mean?). If published, this will include your full peer review and any attached files.

Reviewer #2: No

---

## [Editor Report · Acceptance letter]

29 Sep 2023

PONE-D-23-15200R1 

Maternal constipation is associated with allergic rhinitis in the offspring: A nationwide retrospective cohort study 

Dear Dr. Wei:

I'm pleased to inform you that your manuscript has been deemed suitable for publication in PLOS ONE. Congratulations! Your manuscript is now with our production department. 

Kind regards, 

on behalf of

Dr. Dong Keon Yon 

Academic Editor

PLOS ONE